# Population Genomic Analyses Suggest a Hybrid Origin, Cryptic Sexuality, and Decay of Genes Regulating Seed Development for the Putatively Strictly Asexual *Kingdonia uniflora* (Circaeasteraceae, Ranunculales)

**DOI:** 10.3390/ijms24021451

**Published:** 2023-01-11

**Authors:** Yanxia Sun, Xu Zhang, Aidi Zhang, Jacob B. Landis, Huajie Zhang, Hang Sun, Qiu-Yun (Jenny) Xiang, Hengchang Wang

**Affiliations:** 1CAS Key Laboratory of Plant Germplasm Enhancement and Specialty Agriculture, Wuhan Botanical Garden, Chinese Academy of Sciences, Wuhan 430074, China; 2Center of Conservation Biology, Core Botanical Gardens, Chinese Academy of Sciences, Wuhan 430074, China; 3University of Chinese Academy of Sciences, Beijing 100049, China; 4Center of Economic Botany, Core Botanical Gardens, Chinese Academy of Sciences, Wuhan 430074, China; 5School of Integrative Plant Science, Section of Plant Biology and the L.H. Bailey Hortorium, Cornell University, Ithaca, NY 14853, USA; 6BTI Computational Biology Center, Boyce Thompson Institute, Ithaca, NY 14853, USA; 7Key Laboratory for Plant Diversity and Biogeography of East Asia, Kunming Institute of Botany, Chinese Academy of Sciences, Kunming 650201, China; 8Department of Plant and Microbial Biology, North Carolina State University, Raleigh, NC 27695, USA

**Keywords:** asexuality, allelic heterozygosity, hybrid origin, seed development

## Abstract

Asexual lineages are perceived to be short-lived on evolutionary timescales. Hence, reports for exceptional cases of putative ‘ancient asexuals’ usually raise questions about the persistence of such species. So far, there have been few studies to solve the mystery in plants. The monotypic *Kingdonia* dating to the early Eocene, contains only *K. uniflora* that has no known definitive evidence for sexual reproduction nor records for having congeneric sexual species, raising the possibility that the species has persisted under strict asexuality for a long period of time. Here, we analyze whole genome polymorphism and divergence in *K. uniflora*. Our results show that *K. uniflora* is characterized by high allelic heterozygosity and elevated *π*_N_/*π*_S_ ratio, in line with theoretical expectations under asexual evolution. Allele frequency spectrum analysis reveals the origin of asexuality in *K. uniflora* occurred prior to lineage differentiation of the species. Although divergence within *K. uniflora* individuals exceeds that between populations, the topologies of the two haplotype trees, however, fail to match each other, indicating long-term asexuality is unlikely to account for the high allele divergence and *K. uniflora* may have a recent hybrid origin. Phi-test shows a statistical probability of recombination for the conflicting phylogenetic signals revealed by the split network, suggesting *K. uniflora* engages in undetected sexual reproduction. Detection of elevated genetic differentiation and premature stop codons (in some populations) in genes regulating seed development indicates mutational degradation of sexuality-specific genes in *K. uniflora*. This study unfolds the origin and persistence mechanism of a plant lineage that has been known to reproduce asexually and presents the genomic consequences of lack of sexuality.

## 1. Introduction

Sexual and asexual reproduction are the two basic models of plant reproduction. Although sexual reproduction is the predominant mode in vascular plants, asexual reproduction occurs in many taxonomic groups [1]. Compared to sexual reproduction, asexual reproduction is often described as “a short cut” and “cost effective”, requiring no waiting time and resources for fertilization to occur, resulting in production of more offspring in less time [2]. However, evolutionary theory predicts that asexual reproduction is not a successful long-term strategy. In the absence of sexual reproduction, accumulation of deleterious mutations, e.g., elevated ratio of non-synonymous (selected) to synonymous (neutral) nucleotide diversity (*π*_N_/*π*_S_) is expected due to reduced efficacy of purifying selection in asexual lineages [3,4]. One predicted consequence is the eventual extinction of asexual lineages once the accumulation reaches a high load of deleterious mutations (e.g., Muller’s ratchet) [5,6]. Hence, asexual lineages are traditionally perceived as evolutionary dead-ends [7,8]. However, a number of exceptions, the so-called ‘ancient asexual’ species, e.g., darwinulid ostracods and parthenogenetic oribatid mites, have been reported [9,10,11]. These lineages are suggested to have persisted under obligate asexuality over millions of years, which violates the expectation that sexual reproduction and recombination are necessary for long-term persistence [12]. A long-standing hypothesis is that such ‘ancient asexual’ lineages should have special adaptive mechanisms, e.g., an efficient DNA repair system, to cope with the accumulation of deleterious mutations, e.g., [13,14]. Alternatively, such lineages may not be true ‘ancient asexuals’. Several species (including the famous Bdelloid rotifers) previously believed to be long-term asexuals were indeed later shown to be either recently derived or to engage in some cryptic sexual reproduction [12,15,16,17,18,19].

In the literature, the determination of obligate asexual reproduction in such ‘ancient asexual’ lineages has usually relied on negative evidence, such as the failure to find individuals of the opposite sex for mating or failure to detect a recent sexual ancestor [20]. However, the apparent “obligate” asexuality might reflect our inability to observe sexual reproduction, making dominantly asexual lineages appear strictly asexual [12]. Hence, based on such negative evidence we cannot conclude that sexual reproduction is absent due to the possibility that cryptic sexual reproduction may exist. Assessing the true mode of reproduction in putative ‘obligate asexual’ species requires more reliable methods. There are a number of expected genetic consequences under obligate asexual reproduction. For example, in asexual diploid species, alleles are expected to be highly divergent due to long-term independent accumulation of mutations in the absence of segregation and genetic exchange, the well-known Meselson effect [21,22], or, if the species is of a hybrid nature, due to the divergence of gene copies derived from the different progenitor species [23,24]. Correspondingly, a negative value of the *F*_IS_ index, which measures the level of within-individual heterozygosity, is expected in these lineages [4,25]. Another well-acknowledged signature of asexual reproduction is the generation of non-random associations between loci, i.e., linkage disequilibrium (LD), which is often used for estimating the amount of asexual reproduction. Therefore, genome-wide linkage disequilibrium is expected in obligate asexuality [26,27,28]. In a strictly asexual lineage, where mutation is the only source of genetic novelty and all loci show complete linkage, the genealogy of genes and genomes of the lineage are expected to be a strictly branching tree rather than a network [29]. Additionally, in asexual lineages, genes required specifically for sexual reproduction are expected to be mutated or lost [30]. Although some of the genetic consequences related to asexual reproduction have been assessed in animals, evaluating these predicted consequences has been largely lacking in plant lineages appearing to be asexual only.

*Kingdonia*, represented by a single species *K. uniflora* Balf. f. et W. W. Smith, is one of two monotypic genera (*Kingdonia* and *Circaeaster*) in the family Circaeasteraceae (Ranunculales) [31]. The genus represents an ancient lineage estimated to have diverged from its sister *Circaeaster agrestis* Maxim. during the early Eocene based on molecular dating using the DNA sequences of three chloroplast spacers (52 mya; 95% HPD = 27–75 mya), 497 single-copy genes (51.8 mya; 95% HPD = 31–76 mya), and whole plastome sequences (52.2 mya; 95% HPD = 26–83 mya), respectively [32,33,34]. *Kingdonia uniflora* (diploid, 2n = 2x = 18) is a herbaceous species with a genome size of ~1 Gb, endemic to alpine regions of southwest China and grows in cold and humid habitats with deep humus [33]. The species has a very narrow distribution being restricted to the Qinling Mountains, Minshan Mountains and Daxue-Qionglai Mountains (Figure 1a). Notably, *K. uniflora* is well-known to produce new “individuals” by means of rhizome rupture, which occurs on rhizomes more than three years old [35,36] (Appendix A). Although production of seeds was observed occasionally, so far, field investigations throughout the species’ range found no seedlings of the species in natural populations [36,37]. Efforts to germinate *K. uniflora* seeds in the natural habitat and lab have had a zero rate of success [38]. Additionally, artificially controlled experiments showed that *K. uniflora* does not have apomictic reproduction [39]. Available findings raise the possibility that the species may have evolved without sexual reproduction for a long period of time. Here, we characterize genome-wide genetic variation across nearly all known *K. uniflora* populations. Specifically, we aim to determine (1) if the species is indeed a strictly asexual lineage or cryptic sexual reproduction is present, (2) when asexual reproduction has evolved in the species, and (3) if there are any signs of decay of genes exclusively required for sexual reproduction in this species. To answer these questions and better understand the evolutionary mechanisms of the species, we conducted various analyses to (a) characterize the genetic variation and population structure, (b) examine the genomic signatures of reproductive strategies, (c) infer historical distributions, and (d) identify and annotate genes showing elevated genetic differentiation and containing premature stop codons.

## 2. Results

### 2.1. Sequence Data Processing

We produced 1601 Gb of data containing 2,356,880,312 raw reads for 60 individuals of *K. uniflora*, and 2,342,304,596 clean reads after filtering. The depth of coverage for the 60 samples ranged from 21.6× (WZ4) to 34.8× (TN1), with a mean coverage of 26.6× (Appendix A). Using the *K. uniflora* reference genome (GenBank: PRJNA587615), we obtained 9,349,354 raw SNPs, and 8,678,779 high quality SNPs after primary standard strict filtering. Additionally, by applying deep filtering standards, we obtained 114,746 SNPs located in 960 contigs (Appendix A).

### 2.2. Genetic Diversity and Population Structure

Our LEA (Landscape and Ecological Association) analysis with the cross-entropy method failed to identify an ideal best-fit number of *K* (Appendix A). However, our phylogenetic analysis using the SVDQuartets method [40] revealed three clusters (supported by 74% of all the quartets), corresponding to the QL (Qingling Mountains), MS (Minshan Mountains) and DQ (Daxue-Qionglai Mountains) groups (Figure 1c). Therefore, we repeated the LEA analysis (Figure 1b; Appendix A) by setting *K* = 3. The result recognized the same three groups identified by SVDQuartets. Additionally, two of the three populations from the QL group, three of the five populations from the MS group, and two of the four populations from the DQ group exhibit different levels of genetic admixture with the other groups (Figure 1a,b).

Nucleotide diversity (π) (the average π value) or per-site heterozygosity analysis showed that the three groups have similar genetic diversity, ranging from 1.359 × 10^−3^ to 1.497 × 10^−3^ (Figure 1d). The genetic differentiation statistics (fixation index; *F*_ST_) among the three groups estimated from VCFtools v0.1.16 [41] was 0.136 between the non-adjacent QL and the DQ groups (highest), 0.086 between the adjacent MS group and DQ group (lowest), and 0.104 between the adjacent groups MS and QL (Figure 1d). The Mantel test showed significant isolation by distance (IBD) (*r* = 0.468, *p* = 0.0002) (Figure 1e). Results from the AMOVA showed small genetic variation among the three groups (5.06%; Table 1). The results also showed no significant amount of variation among individuals within populations (with a negative component value of −48%; Table 2). Additionally, most of the genetic variation in *K. uniflora* was explained by variation within individuals (with a component value of 123.83% within individuals) (Table 1), indicating the divergence within *K. uniflora* individuals is large and exceeds the divergence between populations. 

### 2.3. Detecting Predicted Genomic Signatures of Asexuality and Signs of Sexual Reproduction

The calculation of *F*_IS_ using VCFtools v0.1.16 [41] revealed negative *F*_IS_ values (ranging from −0.07 to −0.47) for all 60 *K. uniflora* individuals (mean *F*_IS_ = −0.26) (Figure 2a), indicating greater observed individual heterozygosity than expected from random mating. Both the site frequency spectrum (SFS) analysis based on 12 and 60 individuals showed that sites with heterozygous SNPs shared among three lineages/all 12 populations were most abundant, more frequent than expected under Hardy–Weinberg equilibrium (HWE) (Figure 2b; Appendix A), suggesting a large proportion of the observed excess individual heterozygosity in the 12 populations occurred prior to the evolutionary divergence of the three genetic lineages (QL, MS and DQ). This indicated *K. uniflora* had undergone long-term asexuality (under which individual heterozygosity tends to increase due to the independent evolution of haplotypes) or experienced a hybrid origin of asexuality prior to the evolutionary divergence of the three *K. uniflora* lineages. To discriminate between these two alternatives, we conducted phylogenetic analyses based on phased haplotype sequences. If long-term asexuality accounts for the high individual heterozygosity, the topologies of the two haplotype trees are expected to match each other due to parallel divergence of haplotypes [42]. While our results failed to detect such parallel divergence between haplotype A and B (Figure 3).

Result from Linkage Disequilibrium (LD) analyses showed sites within 0–25 kb region had an average *r*^2^ of 0.60, 0.50 and 0.50, which slowly decay to *c*. 0.50, 0.40 and 0.40 after a physical distance of 300 kb in QL, MS and DQ, respectively (Figure 4a), implying very slow LD decay over very large distance. In addition, the LD analysis between pairs of SNPs up to 1000 kb obtained similar results (Appendix A).

We used the ratio of nonsynonymous to synonymous polymorphisms (*π*_N_/*π*_S_) to assess the efficiency of purifying selection in *K. uniflora*. We defined a total of 24 genotypic groups according to multidimensional scaling (MDS) clustering (Figure 4b). Among the 24 genotypic groups, genetic diversity at zero-fold degenerate (nonsynonymous) and four-fold degenerate (synonymous) sites (*π*_0_ and *π*_4_) is 0.000546 and 0.000992, respectively, resulting in *π*_0_/*π*_4_ (*π*_N_/*π*_S_) = 0.55 (Figure 4c).

The phylogenetic network featured parallel branches (Figure 5), indicating incompatible phylogenetic signals among SNP sites. To determine if the detected conflicting signals are due to recombination, we calculated the PHI test. The result indicated statistically significant signals of recombination (*p <* 0.01).

### 2.4. Detecting and Annotating Genes Showing Elevated Genetic Differentiation and Containing Premature Stop Codons

The average *F*_ST_ between populations was estimated as 0.24 but ranged from 0.05 to 0.76 (Appendix A). BAYESCAN identified 7917 outlier (*F*_ST_ values ranging from 0.47 to 0.76) sites out of the total 114,746 SNPs. Through comparisons with the reference genome, we found these outliers were located in 159 genes, of which 87 had BLAST hits in the Unified Protein Database, related to 15 biological processes (second-level gene ontology (GO) terms) (Figure 6a). Gene set enrichment (GSE) analysis for all 87 genes detected five significantly over-represented terms (*p* < 0.05) associated with reproduction, including: “reproduction” (13 genes), “reproductive process” (13 genes), “reproductive developmental process” (11 genes), “fruit development” (6 genes), and “seed development” (6 genes) (Figure 6b; Table 2). By analyzing the sequences of these enriched genes, we detected premature stop codons caused by single-nucleotide variation (in some individuals) in four of the six genes related with seed development (Appendix A).

## 3. Discussion

### 3.1. Kingdonia uniflora Is Characterized by High Allelic Heterozygosity, Slow Linkage Disequilibrium Decay and Elevated π_N_/π_S_ Ratio, and Shows Signs of DNA Recombination

In diploid asexuals, high levels of allelic divergence are expected to result from two factors: (1) long-term evolution under obligate asexuality, i.e., Meselson effect [21,22] and (2) transition to asexual reproduction via hybridization between sexual species [24]. The Meselson effect is usually considered to be a strong indicator of long-term evolution under strict asexuality [11,44], yet this phenomenon has been shown to appear in relatively young lineages of less than 100,000 years age [45]. Here, we detected high allelic heterozygosity in the flowering plant species *K. uniflora*: (1) an excess of observed individual heterozygosity over Hardy–Weinberg expectation as indicated by negative *F*_IS_ values (which is different from what is observed in the sister sexual species *C. agrestis* (mean *F*_IS_ = 0.02; Table 1 in [34]) and (2) greater genetic divergence within individuals than that between populations revealed by the AMOVA analysis. Theoretically, as mentioned above, both the Meselson effect and hybrid origin can explain high allelic heterozygosity in diploid asexuals. While the topologies of the two haplotype trees are expected to match each other due to their parallel divergence in the case of Meselson effect. Our results from phylogenetic analyses based on phased haplotype sequences failed to detected such mirrored topologies, indicating long-term asexuality may not be the cause of high allelic heterozygosity in *K. uniflora*. In addition, we detected recombination signs in the genealogical network, suggesting possible occurrence of sexual reproduction. The evidence supports that *K. uniflora* is likely not an obligate asexual species. Although the detected recombination events could have been from mitotic recombination, due to the following evidence, we argue that occasional successful sexual reproduction in the species is more likely the cause. First, the species still produces seeds, although no seedling has been observed in the field [37,46]. Second, high allele divergence has been shown to be compatible with low-rate sexual reproduction [47]. Third, the LEA analysis revealed that some individuals exhibit genetic admixture. Lastly, SVDQuartets showed a quarter of the quartets were incongruent with the species tree, indicating a portion of the SNPs did not diverge congruently with the rest of the SNP sites due to incomplete lineage sorting or recombination. Therefore, we hypothesize that the species likely engages, to some extent, in sexual reproduction. The maintenance of high heterozygosity in the case of occurrence of DNA recombination further points towards a hybrid origin for *K. uniflora*. Hence, *K. uniflora* likely switched to asexuality via interspecific hybridization, as reported in asexual species of *Meloidogyne*, *Lineus* ribbon worms, and the *Ranunculus auricomus* complex [4,15,45,48]. Different from the above hybrid origins of asexual species with multiple congeneric sister species, *Kingdonia* is a monotypic genus and has no fossil record that indicates the presence of parental species for hybridization. The closest related species, *C. agrestis*, is from a different genus and diverged from *K. uniflora* tens of millions of years ago [33,34]. Nevertheless, we cannot exclude the possibility that these congeneric species of *K. uniflora* did in fact exist but were simply not distinguishable in the fossil record or were just not preserved. The site frequency spectrum reveals the level of the heterozygous genotypes shared among lineages greatly exceed those under Hardy–Weinberg equilibrium, indicating high allele divergence in *K. uniflora* occurred prior to lineage differentiation. This supports that *K. uniflora* had switch to asexuality (probably via interspecific hybridization) before lineage differentiation. 

Asexual reproduction prevents free exchange of alleles among individuals and results in allele linkage disequilibrium (LD) and in extreme cases (obligate asexuality) may result in complete physical linkage of markers over the entire genome [12,49,50]. Various genetic processes in outcrossing can reduce LD and free random mating results in linkage equilibrium of alleles. Therefore, the level of LD in a species reflects the extent of inbreeding (non-random mating) or asexuality. The pattern of LD decay in *K. uniflora* is comparable to that seen in the highly self-fertilizing/asexual plant species, e.g., *Arabidopsis thaliana* [51,52], *Medicago truncatula* [53], and *Spirodela polyrhiza* [54], and completely different from outcrossing populations that often show rapid LD decay over several hundred bp [54,55,56]. The LD decay pattern provides genetic evidence supporting *K. uniflora* as a species is undergoing reproduction by high asexuality.

Interference among loci caused by high linkage disequilibrium will decrease the efficiency of selection by preventing selection from acting individually on each locus [57,58]. Such kinds of selective interference can result in a higher ratio of non-synonymous (selected) to synonymous (neutral) polymorphisms in asexual lineages [4,44]. The value of *π*_N_/*π*_S_ ratio (0.55) in *K. uniflora* is relatively higher than that (ranging from 0.20 to 0.35) observed in outcrossing plants [43], implying a higher rate of non-synonymous substitution caused by reduced efficacy of purifying selection in *K. uniflora*. Under high levels of asexuality, mutations are expected to be an important source of variation, with mutations typically occurring in a heterozygous state in asexual species due to independent evolution of alleles. However, recessive mutations are not exposed to selection [4], which could be another reason why a higher rate of non-synonymous substitution is detected in *K. uniflora*.

### 3.2. Genes Regulating Seed Development Show Signs of Decay

Theoretically, local adaptation involving diversifying selection could maintain and enhance genetic polymorphisms, but this adaptation is constrained in asexual lineages due to genome-wide interference among loci [28,59]. Our results showed that genes playing a strong role in genetic differentiation among *K. uniflora* populations are enriched for functions involved in seed development. The fact that field surveys and lab experiments failed to observe successful seed germination indicates failure or low rate of seed development and germination, which is likely the major factor contributing to non-evident sexual reproduction in *K. uniflora*. The differences among populations in environmental variables, e.g., temperature and precipitation, may directly affect seed development and germination rates among populations, further driving diversifying selection in shaping the diversity of corresponding genes. However, given divergent selection might be constrained in *K. uniflora* due to high linkage disequilibrium detected in the species, significant genetic differentiation of genes regulating seed development in *K. uniflora* may be signs of mutational degradation of genes associated with sexual reproduction. Especially detection of premature stop codons in the genes regulating seed development in some populations further indicates a sign of decay of genes specific to sexual reproduction in *K. uniflora*. Theoretically, for asexual lineages, mutation or loss of genes required exclusively for sexual reproduction is expected due to relaxed selection on these genes [30]. The loss of four of the eight genes important for meiosis is indeed detected in the *K. uniflora* genome [33], in line with the above theoretical expectation. Hence, it is quite possible that the decay of genes regulating seed development in *K. uniflora* is closely related with the lack of successful sexual reproduction in this species.

## 4. Methods and Materials

### 4.1. Sample Collection and Resequencing

We carefully selected a total of 12 populations (Appendix A) covering the whole distribution range of *K. uniflora*. We collected fresh leaves of five individuals that were at least ten meters away from each other in each population to reduce the possibility that each sample was not a physiologically independent ramet. We dried the leaves in silica gel in the field, then stored them at −20 °C before DNA extraction. Genomic DNA extraction, library construction, and amplification followed the protocols of Novogene (Beijing, China) (Appendix A). All samples were sequenced using the Illumina HiSeq 4000 platform with a pair-end read length of 150 bp by Novogene Bioinformatics Technology Co., Ltd. (Beijing, China). Illumina raw reads were filtered by removing adapters and low-quality reads using Trim Galore v0.6.5 (https://www.bioinformatics.babraham.ac.uk/projects/trim_galore (accessed on 1 January 2021)) with default options (Phred quality threshold 20; adapter auto-detection).

### 4.2. SNP Calling and Quality Control

We mapped the filtered reads of each individual to the *K. uniflora* genome [33] using BWA-MEM v0.7.12-r1039 [60] with default parameters. We then converted sequence alignment/map (SAM) format files to BAM and sorted the BAM files using SAMtools v1.6 [61] and conducted the following analyses in Genome Analysis Toolkit (GATK, v4.0) [62]. We first marked duplicate reads using MarkDuplicates. We then processed the data with AddOrReplaceReadGroups to replace all read groups in the INPUT file with a single new read group name and assign all reads to this read group in the OUTPUT BAM. To identify SNPs, we first conducted single-sample haplotype calling with HaplotypeCaller, and then identified Multi-sample SNPs using GenotypeGVCFs after merging the haplotype caller results from each sample using CombineGVCFs which aligned the haplotypes among samples. To obtain high quality SNPs, we filtered multi-sample SNPs using VariantFiltration with strict filter settings “QD (QualByDepth) < 2.0 || MQ (RMSMappingQuality) < 40.0 || FS (FisherStrand) > 60.0 || SOR (StrandOddsRatio) > 3.0 || MQRankSum (MappingQualityRankSumTest) < −12.5 || ReadPosRankSum (ReadPosRankSumTest) < −8.0”. We further filtered the SNPs data to exclude monomorphic or triallelic variants, indels, and SNPs missing in any samples via VCFtools v0.1.16 [41] for further analyses.

### 4.3. Genetic Diversity and Population Structure

To assess the population structure of *K. uniflora*, we performed the following analyses. We used Landscape and Ecological Association (LEA) (v3.3.2) R package [63] to determine the number of ancestral populations. LEA was developed for large genotypic matrices and does not rely on the genetic assumptions of the absence of genetic drift, Hardy–Weinberg or linkage equilibrium in ancestral populations [64]. We also used a coalescent-based method to infer a “species tree” based on the SNP data for comparison with results from LEA. Specifically, we used the SVDQuartets method [40] implemented in PAUP* v4.0a166 (http://paup.phylosolutions.com/ (accessed on 1 January 2021)) with 100 bootstrap replicates and the quartet assembly method QFM to produce a species tree [65]. 

We estimated nucleotide diversity (π) and genetic differentiation (*F*_ST_) between groups identified in the aforementioned analyses using VCFtools v0.1.16 [41], and used Analysis of Molecular Variance (AMOVA) in Arlequin v3.5.2.2 [66] to estimate relative contributions of genetic variation from within and between groups. To determine if genetic differentiation is associated with geographic distance, we performed an IBD (isolation by geographic distance) analysis. We first estimated population-level genetic differentiation *F*_ST_ using the Weir and Cockerham method [67] implemented in HIERFSTAT [68] in R v3.6.1 [69]. We then calculated genetic distance with the formula *F*_ST_/(1 − *F*_ST_) and computed the pairwise geographic distance among 12 populations using GENALEX v6.5 [70]. We then tested the significance for the relationship between geographical distances and genetic distance among populations by conducting Mantel tests with ADE4 v1.7 (https://CRAN.R-project.org/package=ade4 (accessed on 1 January 2021)) using *mantel.rtest* with 9999 permutations.

### 4.4. Detecting Predicted Genomic Signatures of Asexuality and Signs of Sexual Reproduction

To understand the reproductive strategies, we first employed VCFtools v0.1.16 [41] (option-het) to calculate *F*_IS_ = 1 − *H*_obs_/*H*_exp_ for each individual, where *H*_obs_ and *H*_exp_ are the observed and expected heterozygosity, respectively. A negative *F*_IS_ value indicates an excess of observed individual heterozygosity. We then examined the distribution pattern of the excessive heterozygosity among the genetic groups identified in the aforementioned analyses by generating a site frequency spectrum (SFS) using Pop-Con with standard parameters (https://github.com/YoannAnselmetti/Pop-Con (accessed on 1 December 2021)). All 60 individuals and 12 randomly selected individuals that cover all populations were used for the SFS analysis, respectively.

We further conducted phylogenetic analyses based on phased haplotype sequences. The SNPs filtered by VCFtools were phased using BEAGLE v5.2 [71]. Then, output data from BEAGLE were converted into haplotype sequences for the corresponding positions in the reference genome using a perl script (available as Appendix A: GetSnpSeq.pl). Pairwise distance of each haplotype to the reference genome was calculated using snp-dist (https://github.com/tseemann/snp-dists (accessed on 1 January 2021)) to distinguish haplotype A from haplotype B (haplotype A being closer to the reference genome, haplotype B being more diverged) [11]. Phylogenetic analyses based on phased haplotype A and B sequences were conducted using the neighbor-joining (NJ) method implemented in MEGA (v10.1.6) [72] with default parameters, respectively. 

We then examined other predicted genetic consequences of asexual reproduction (i.e., linkage disequilibrium (LD) and reduced efficiency of purifying selection). We calculated pairwise linkage disequilibrium (*r*^2^ value) and modeled the decline of LD with physical distance using PopLDdecay v3.40 [73] with default settings. Within each contig we calculated LD between pairs of sites up to 300 kb and 1000 kb, respectively. In obligate asexuals, genome-wide LD between loci is expected, and the decline of LD between loci does not depend on their physical distance [74]. 

To test if *K. uniflora* is characterized by reduced efficacy of purifying selection, we calculated the *π*_N_/*π*_S_ ratio of the species. For asexual species, individuals from the same population may be clones via descending from a common ancestor via only clonal reproduction, possibly over many generations [54]. Therefore, prior to the calculation, we grouped individuals by conducting multidimensional scaling (MDS) analysis in plink v1.9 with the options -cluster, -mds-plot 2 eigvals and -allow-extrachr [75] to group individuals that were genotypically highly similar into the same genotypic group within a population. We then randomly selected one individual from each genotypic group and generated a concatenated coding sequence for each selected individual based on SNP loci from coding regions using a custom script (Appendix A). The sequence matrix comprising all above selected individuals was used for following nucleotide diversity calculation. Designated nucleotide diversity at 0-fold and 4-fold degenerate positions, *π*_N_ (*π*_0_) and *π*_S_ (*π*_4_), were calculated in MEGA (v10.1.6) [72].

To assess if any potential cryptic sexual reproduction may have occurred in *K. uniflora*, we reconstructed the relationship of individuals using SNP data with the NeighborNet method in SplitsTree v4.13.1 [76] with default settings. The method does not force a tree-like phylogeny in the analysis and can reveal phylogenetic networks resulting from divergent phylogenetic signals. We further ran the Pairwise Homoplasy Index (PHI) test for recombination using SplitsTree v4.13.1 [76].

### 4.5. Detecting and Annotating Genes Showing Elevated Genetic Differentiation and Containing Premature Stop Codons

To determine if there are any signs of mutational degradation of genes exclusively required for sexual reproduction in *K. uniflora*, we conducted following analyses. First, we identified SNPs displaying higher levels of divergence than expected under neutrality by performing an overall *F*_ST_ outliers test in BAYESCAN v2.1 [77] with default parameters. The locus-specific component (*α*) was used to identify loci whose *F*st values were larger than expected from coalescent simulation of neutral evolution (*α* > 0). Significance is based on FDR-corrected *q*-values (<0.01). Second, we determined what functions the genes with SNPs showing elevated genetic differentiation may have by annotating each of these genes using gene ontology (GO) terms with TBtools [78]. We then used the Singular Enrichment Analysis (SEA) tool in agriGO v2.0 [79] to analyze gene enrichment and tested for statistical significance of gene enrichment using the chi-squared test. Third, we checked for the existence of premature stop codons in the enriched genes by employing BioEdit v7.0.9.0 [80].

## 5. Conclusions

Old asexual lineages are of particular interest from the perspective that they seem to defy expectations regarding rapid extinction in the total absence of sexual reproduction, providing an opportunity to use these species to study the evolutionary consequences of asexuality. Multiple such putative examples exist in the animal kingdom (though they typically end up losing their “ancient asexual” status once occasional sexuality is discovered), the potential for ancient asexuality is much less well characterized in plants. *Kingdonia uniflora* represents a fascinating natural system for population genomic analyses to address the question in evolutionary biology about the origin, persistence and fate of asexuality in plants. In this study we use genome-wide SNP markers to survey nearly all known populations of *K. uniflora.* We conducted a series of analyses to infer the population structure, the level of individual heterozygosity and linkage disequilibrium, the efficacy of purifying selection, and the association between genetic variations and asexual reproduction. In conclusion, our results suggest the putatively strictly asexual *K. uniflora* has a hybrid origin and engage in some extent sexual reproduction; the genes related with seed development show signals of mutational degradation in the species.

## Figures and Tables

**Figure 1 ijms-24-01451-f001:**
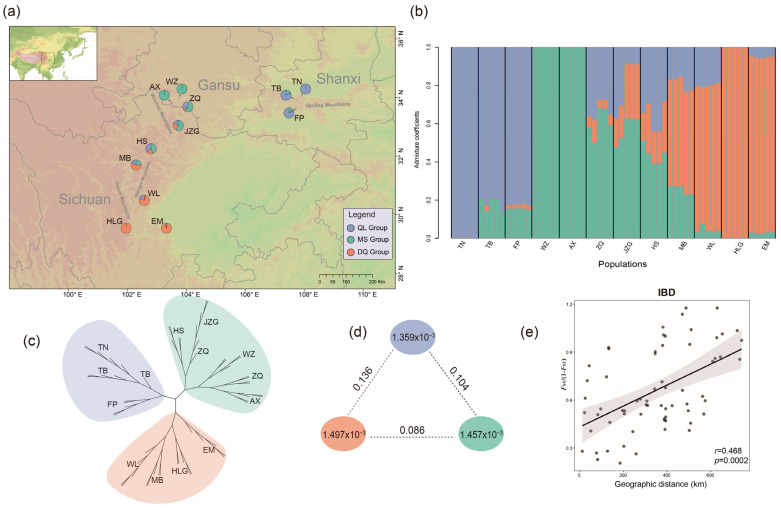
Genetic structure and diversity within *K. uniflora*. (**a**) Sampling sites and genetic structure detected by LEA analysis (*K* = 3 populations) mapped using ArcGIS v10.3. The genetic QL group is shown in blue, the MS group is shown in green, and the DQ group is shown in orange. (**b**) Genetic groups of *K. uniflora* inferred by the LEA analysis when setting *K* = 3. (**c**) A phylogeny derived from SVDQuartets using 114,746 SNPs. (**d**) Nucleotide diversity (π) and population divergence (*F*_ST_) across three genetic groups. The value in each circle represents a measure of nucleotide diversity for this group, and the value on each line indicates divergence between groups. (**e**) Correlation between genetic distance and geographical distance (isolation by distance, IBD), as tested by Mantel test.

**Figure 2 ijms-24-01451-f002:**
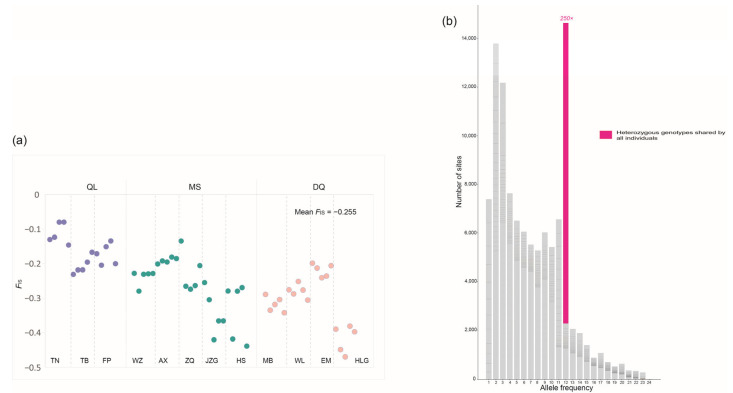
Analyses of individual heterozygosity in *K. uniflora*. (**a**) Distribution of values of inbreeding coefficient (*F*_IS_). Under Hardy–Weinberg equilibrium *F*_IS_ is 0, negative values of *F*_IS_ indicate excess of individual heterozygosity. (**b**) The site frequency spectrum (SFS) depicting the number of sites with different allele frequency across 12 *K. uniflora* individuals (one allele can display a maximum frequency of 24 among 12 diploid individuals). Heterozygous genotypes shared among all 12 populations are highlighted using the purple color and its excess over HWE indicated (*c*. 250 times as frequent as expected under HWE).

**Figure 3 ijms-24-01451-f003:**
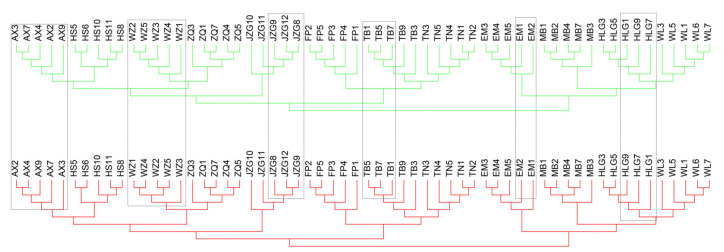
Haplotype phylogenies observed in *K. uniflora* based on 114,746 SNPs. The phylogenetic trees constructed based on phased haplotype A and B sequences are highlighted with red (haplotype A) and green (haplotype B) color, respectively. The differences between the two tree topologies are marked using rectangles with dashed lines.

**Figure 4 ijms-24-01451-f004:**
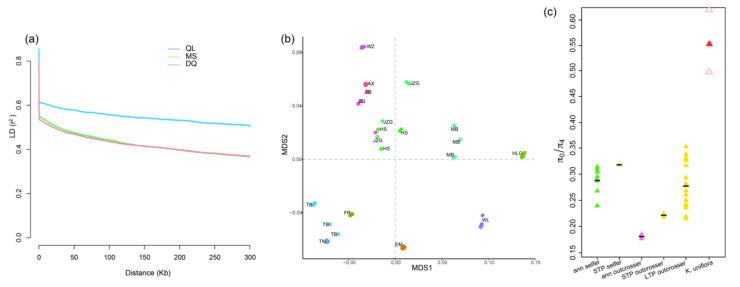
Analyses of genetic signatures expected under asexuality. (**a**) Decay of linkage disequilibrium (LD) with physical distance in three genetic groups of *Kingdonia uniflora*. Averages of pairwise linkage disequilibrium measures *r*^2^ are plotted for each bin of distances between pairs of SNPs. The displayed data are for the bins with pairs of SNPs separated by ≤300 kb. (**b**) Multidimensional scaling analysis of identity-by-state pairwise distances between individuals. Individuals in the same population are coded with the same color. Individuals that come from the same population and cluster together are grouped into one genotypic group, and a total of 24 genotypic groups are identified. (**c**) *π*_0_/*π*_4_ difference between *K. uniflora* and other plants reported in Chen et al. (2017) [43]. The figure is modified based on Figure 2B in Chen et al. (2017) [43]. STP, short-term perennial; LTP, long-term perennial. The red triangle shows the average value of *π*_0_/*π*_4_ in *K. uniflora*; the triangles with dashed lines show the range of *π*_0_/*π*_4_ values in *K. uniflora*.

**Figure 5 ijms-24-01451-f005:**
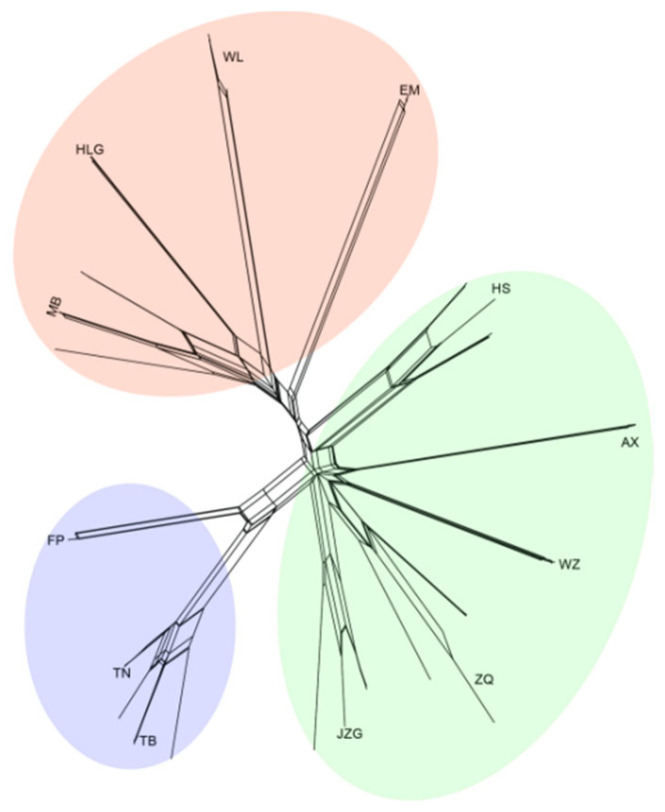
The SplitsTree phylogenetic network constructed using a total of 114,476 SNPs. Parallel edges represent the splits computed from the data. The clusters corresponding to different genetic groups are shown with different colors. The genetic QL group is shown in blue, the MS group is shown in green, and the DQ group is shown in orange.

**Figure 6 ijms-24-01451-f006:**
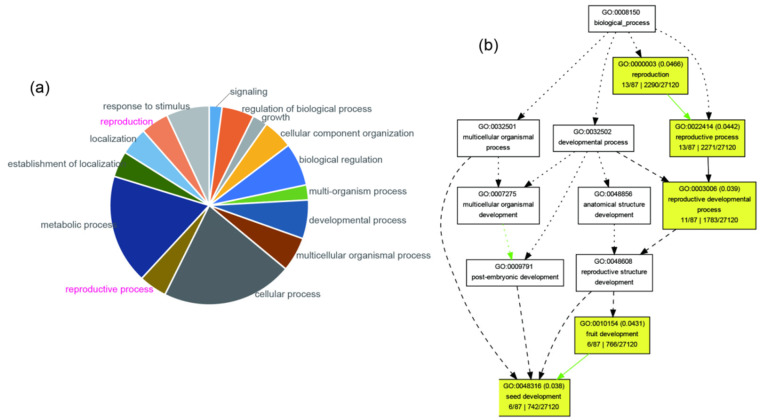
Analyses of genes showing elevated differentiation. (**a**) Pie chart of second-level gene ontology (GO) terms (biological process) of genes showing outlier *F*st values. (**b**) Graphical results showing enriched GO terms (second to sixth level; highlighted with yellow color) of genes containing outlier SNPs.

**Table 1 ijms-24-01451-t001:** Details on the results of the analysis of the molecular variance (AMOVA).

Source of Variation	Sum of Squares	Variance Components	Percentage Variation	Fixation Indices	*p*-Values
Among groups	166,030.43	993.35	5.06	0.05	**0.00**
Among populations within groups	396,979.34	3879.21	19.76	0.21	**0.00**
Among individuals within populations	255,202.00	−9443.63	−48.10	−0.64	1.00
Within individuals	1,452,238.00	24,203.97	123.28	−0.23	1.00

Bold values indicate *p* < 0.05.

**Table 2 ijms-24-01451-t002:** The significantly over-represented GO terms of genes containing outlier SNPs in *K. uniflora*.

GO Term	Class	Description	Significant	Annotated	*p*-Value
GO:0048316	P	seed development	6	742	0.038
GO:0003006	P	reproductive developmental process	11	1783	0.039
GO:0010154	P	fruit development	6	766	0.043
GO:0022414	P	reproductive process	13	2271	0.044
GO:0000003	P	reproduction	13	2290	0.047

## Data Availability

*K. uniflora* resequencing reads have been deposited at the SRA repository under bioproject number PRJNA611722.

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
