# Peer review of "Population Genomic Analyses Suggest a Hybrid Origin, Cryptic Sexuality, and Decay of Genes Regulating Seed Development for the Putatively Strictly Asexual Kingdonia uniflora (Circaeasteraceae, Ranunculales)"

_ijms, 2023, doi:10.3390/ijms24021451_

Round 1

Reviewer 1 Report

The morphology of the plant can be discussed with current molecular findings.

Author Response

Reviewer 1

The morphology of the plant can be discussed with current molecular findings

Thanks so much for the suggestion. But we failed to detect any evident correlations between current molecular findings and K. uniflora morphologies. But in our another work (unpublished), by conducting comparative genomic analyses between K. uniflora and its sister species Circaeaster agrestis, we found a significant association between well-developed rhizome of K. uniflora and obvious expansion of gene families regulating root system development in this species.

Reviewer 2 Report

This study presents genomic analysis of an asexual plant lineage that provides insights into the age of the lineage, its formation, and the consequences of long-term asexuality. The abstract summarises the paper well, emphasizing the interest in the system as a rare plant example of a long-term asexual lineage and the variety of interesting insights gained from genomic analysis. The introduction provides strong wider context about the evolution of asexuality that generates wider interest in this study system. A review of the literature leads to three clearly stated aims for the paper. The methods are concisely and clearly explained. The supplemental files with perl code are a welcome record to assist other researchers with similar analyses. The species distribution modelling seems less linked to the study's main aims to investigate asexuality. Perhaps this part needs further justification at the end of the introduction or consider removing it from the paper to simplify the message. The results are also concise and figures are clear with informative legends. The results include some analyses such as phased haplotypes that are missing from the methods. It is important to include these as they address important questions about testing the hybrid origin of the lineage. Figure S5 might be worth including in the main paper as it addresses this hybrid origin hypothesis. While the A and B phylogenies are not strictly congruent, they are highly so, which does not completely rule out an ancient asexual origin, particularly if rare sexuality occurs. The results of other analyses such as SplitTrees would also support this perspective. The discussion does well to interpret the results in terms of the study aims and relate it to the wider field. I find the strong conclusion of hybrid origin versus long term asexuality problematic as it does not adequately consider limited sexuality. Limited sexuality is discussed separately but combined discussion of these two aspects to evaluate likely scenarios would be helpful. Overall this is a fascinating study contributing significant new genomic results addressing a evolutionary questions about the origin and persistence of asexuality. The multiple results and interpretations still need some more organising to make a more coherent story but this paper would be a valuable contribution to the field.

Specific comments
L31 Use singular term for "periods"
L37 Name the sequences used to define these haplotypes.
L39 Typo: Replace "statistically" with "statistical"
L95-97 Should this be "maintenance of divergent gene copies derived from the different progenitor species"?
L220-221 Why the need for a subset of individuals for the SFS analysis?
L262-263 What is the argument for checking the past distribution at this timepoint?
L284-285 What tool(s) were used to check for stop codons?
L289 It seems like Genbank submission remains to be confirmed before publication of this article.
L295 Why does supplemental table count start at S4?
L317-319 The term "admixture" seems contradictory when discussing an asexual lineage. I wonder if a line or two of justification for the use of LEA that assumes sexually reproducting populations is needed in the methods.
L353-355 This analysis needs to be described in the main methods. How were haplotypes phased using short reads?
L359-361. These conclusions could be moved to the discussion. Would limited sexual reproduction (not ruling out ancient asexuality) be another viable conclusion?
L408-417 including Figure 5. This analysis still seems poorly linked to the overall study aims and i would consider omitting it from this paper. If kept, mark the current distribution/sample sites on these maps
L430-433 Were stop codons detected in any of the other genes grouped by other functions?
Figure 6a This an interesting visualisation of the Fst outlier analysis but i am not sure it is very helpful. The ordering of SNPS is not very useful if the reference genome still consists of many scaffolds. Further, the figure draws attention to a region of low Fst that is not addressed in the paper. Better to omit this figure in my opinion.
L446-447 None of the presented results are currently linked to the interpretation of cryptic sexual reproduction.
L454-455 To me, 100,000 years does not seem very young. Maybe say "relatively young"? Also better to state it as generations to accommodate the diverse examples of asexuality you discuss.
L468-469 This feels like an overly strong conclusion, given that limited sexuality could also explain the slightly incongruent phylogenies.
L501-502 No need for a paragraph break here.
L543-546 One could argue that mountains are dispersal barriers to many species regardless of their reproductive strategy. There are lots of plant examples in the literature.
L546-553 I don't think the term "expansion" applies if the earlier distribution range was larger, especially since this part of the study reflects two snapshots of particular time points. Overall, i feel that this part of the paper is tangential to the main aims.
L554-580 The observed association between high differentiation and mutated genes for sexual reproduction is odd and the authors do well to consider several contrasting possibilities. I wonder if relatively recent post-asexuality gene decay could be added to the list, especially in view of IBD results suggesting limited dispersal across the landscape.
All pages are numbered as 1.

Author Response

We are very grateful to the reviewer for reading our manuscript carefully and giving detailed comments for us to improve the manuscript. We have revised our manuscript according to the reviews. The following is our point-by-point response to the reviewer’s comments.       

This study presents genomic analysis of an asexual plant lineage that provides insights into the age of the lineage, its formation, and the consequences of long-term asexuality. The abstract summarises the paper well, emphasizing the interest in the system as a rare plant example of a long-term asexual lineage and the variety of interesting insights gained from genomic analysis. The introduction provides strong wider context about the evolution of asexuality that generates wider interest in this study system. A review of the literature leads to three clearly stated aims for the paper. The methods are concisely and clearly explained. The supplemental files with perl code are a welcome record to assist other researchers with similar analyses.

The species distribution modelling seems less linked to the study's main aims to investigate asexuality. Perhaps this part needs further justification at the end of the introduction or consider removing it from the paper to simplify the message.

After consideration, we decided to remove the ‘species distribution modelling’ part, and this part has been removed in the revised manuscript.   

The results are also concise and figures are clear with informative legends.

The results include some analyses such as phased haplotypes that are missing from the methods. It is important to include these as they address important questions about testing the hybrid origin of the lineage.

In the revised manuscript, we have moved the methods for ‘Allele tree construction’ from Supplemental_ Notes to the Methods part of the main text.

Figure S5 might be worth including in the main paper as it addresses this hybrid origin hypothesis.

We have moved Figure S5 to the main text, please see Figure 3 in the revised manuscript.

While the A and B phylogenies are not strictly congruent, they are highly so, which does not completely rule out an ancient asexual origin, particularly if rare sexuality occurs.

We totally understand the reviewer’s concern here. Ancient asexuals are defined as those species having persisted under strict asexuality for a long time. For asexual species, high individual heterozygosity could be caused by long-term strict asexuality or a hybrid origin. Failing to detect parallel divergence of haplotypes indicated a signal of a hybrid origin for K. uniflora; in addition, the following detection of DNA recombination events suggested that K. uniflora is not strictly asexual; the maintenance of high heterozygosity in the case of occurrence of DNA recombination further points towards a hybrid origin for K. uniflora

The results of other analyses such as SplitTrees would also support this perspective. The discussion does well to interpret the results in terms of the study aims and relate it to the wider field. I find the strong conclusion of hybrid origin versus long term asexuality problematic as it does not adequately consider limited sexuality. Limited sexuality is discussed separately but combined discussion of these two aspects to evaluate likely scenarios would be helpful.

Done. We have combined these two aspects together to improve our discussion in the revised manuscript. 

Overall this is a fascinating study contributing significant new genomic results addressing a evolutionary questions about the origin and persistence of asexuality. The multiple results and interpretations still need some more organising to make a more coherent story but this paper would be a valuable contribution to the field.

Specific comments
L31 Use singular term for "periods"

Done.

L37 Name the sequences used to define these haplotypes.

The two haplotypes were generated by phasing the SNPs generated in the present study, please also see the Methods part (lines 233-245) in the revised manuscript. 

L39 Typo: Replace "statistically" with "statistical"

Done.

L95-97 Should this be "maintenance of divergent gene copies derived from the different progenitor species"?

Yes.

L220-221 Why the need for a subset of individuals for the SFS analysis?

Please see Supplemental Fig. S4 which includes all 60 individuals for the analysis, it is difficult to read the results, unless you enlarge the image enough. Hence, to display the results clearly, we conducted the analysis using a subset of individuals, and got the same conclusion that sites with heterozygous SNPs shared among three lineages/all 12 populations were most abundant, more frequent than expected under Hardy-Weinberg equilibrium (HWE). 

L262-263 What is the argument for checking the past distribution at this timepoint?

We have deleted this part in the revised manuscript.

L284-285 What tool(s) were used to check for stop codons?

We have added this information in the revised manuscript (line 296).

L289 It seems like Genbank submission remains to be confirmed before publication of this article.

We have rewritten this sentence and provided the related information in the ‘Data availability statement’ part.

L295 Why does supplemental table count start at S4?

Figure S4 is too large, so we separated it from the other supplemental tables which can be found in the file Supplemental_Fig _S1-S3, S5-S6, Table _S1-S3, S5.

L317-319 The term "admixture" seems contradictory when discussing an asexual lineage. I wonder if a line or two of justification for the use of LEA that assumes sexually reproducting populations is needed in the methods.

We have added the information the reviewer suggested in the revised manuscript (lines 194-197).

L353-355 This analysis needs to be described in the main methods. How were haplotypes phased using short reads?

Done

L359-361. These conclusions could be moved to the discussion. Would limited sexual reproduction (not ruling out ancient asexuality) be another viable conclusion?

Ancient asexuality refers to being under strict asexuality for a long time. The maintenance of high heterozygosity in the case of occurrence of sexual reproduction (although it is limited) points even towards a hybrid origin conclusion.

L408-417 including Figure 5. This analysis still seems poorly linked to the overall study aims and i would consider omitting it from this paper. If kept, mark the current distribution/sample sites on these maps

We have deleted this part.

L430-433 Were stop codons detected in any of the other genes grouped by other functions?

If ‘the other genes grouped by other functions’ refers to those listed in Table 2,   the answer is ‘yes’, because the four genes having premature stop codons belong to the ‘seed development’ group, but also belong to the ‘reproduction’ and ‘reproductive progress’ group in Table 2.    

Figure 6a This an interesting visualisation of the Fst outlier analysis but i am not sure it is very helpful. The ordering of SNPS is not very useful if the reference genome still consists of many scaffolds. Further, the figure draws attention to a region of low Fst that is not addressed in the paper. Better to omit this figure in my opinion.

We have moved Figure 6a to the supplemental materials (Figure S6). 

L446-447 None of the presented results are currently linked to the interpretation of cryptic sexual reproduction.

We have reorganized this part in the revised manuscript.

L454-455 To me, 100,000 years does not seem very young. Maybe say "relatively young"? Also better to state it as generations to accommodate the diverse examples of asexuality you discuss.

We have added the word ‘relatively’ in this sentence. There is no information about generations in the original literature.  

L468-469 This feels like an overly strong conclusion, given that limited sexuality could also explain the slightly incongruent phylogenies.

Please see our explanations above, and we have reorganized this part in the revised manuscript.

L501-502 No need for a paragraph break here.

Done.

L543-546 One could argue that mountains are dispersal barriers to many species regardless of their reproductive strategy. There are lots of plant examples in the literature. 546-553 I don't think the term "expansion" applies if the earlier distribution range was larger, especially since this part of the study reflects two snapshots of particular time points. Overall, i feel that this part of the paper is tangential to the main aims.

We have deleted this part in the revised manuscript.

L554-580 The observed association between high differentiation and mutated genes for sexual reproduction is odd and the authors do well to consider several contrasting possibilities. I wonder if relatively recent post-asexuality gene decay could be added to the list, especially in view of IBD results suggesting limited dispersal across the landscape.

As we discussed in this part, we inferred that the decay of the genes (required exclusively for sexual reproduction) occurred after the origin of asexuality in K. unifora, but currently we are not certain whether the decay is recent or ancient. 

All pages are numbered as 1.

We are also confused about this, the version for peer-review was generated by the Editorial Office. But the version we download for revision is normal. 

Reviewer 3 Report

This article submitted for revision (ijms-2056005) presents an in -depth genetic analysis of Kingdonia uniflora genome to better understand the evolutionary mechanisms of  reproduction modes of this species. The results of this experiment revealed that K.uniflora is likely not an obligate asexual species and it engages in undetected sexual reproduction.  The experiment is properly designed and clearly presented. I have only one remark to the text; in line 123 I suggest to make a supplement related to the genetic description  of the species 2n=2x=18. Regarding the extensive population genomic analyses  of Kingdonia uniflora provided via this research I recommend to publish this article in IJMS journal. 

Author Response

Reviewer 3

This article submitted for revision (ijms-2056005) presents an in -depth genetic analysis of Kingdonia uniflora genome to better understand the evolutionary mechanisms of  reproduction modes of this species. The results of this experiment revealed that K.uniflora is likely not an obligate asexual species and it engages in undetected sexual reproduction.  The experiment is properly designed and clearly presented.

I have only one remark to the text; in line 123 I suggest to make a supplement related to the genetic description of the species 2n=2x=18.

Done. We have replaced 2n=18 with 2n=2x=18

Regarding the extensive population genomic analyses  of Kingdonia uniflora provided via this research I recommend to publish this article in IJMS journal.